applied mathematics/mathematical modelling/computational mathematics

uncertainty quantification, transitional Markov chain Monte Carlo, inverse problem, one-dimensional blood flow, model selection

**Author for correspondence:**
Anastasios Matzavinos
e-mail: matzavinos@brown.edu

# Detection of arterial wall abnormalities via Bayesian model selection

Karen Larson[1], Clark Bowman[2], Costas Papadimitriou[3], Petros Koumoutsakos[4] and Anastasios Matzavinos[1]

[1]Division of Applied Mathematics, Brown University, Providence, RI 02912, USA
[2]Department of Mathematics and Statistics, Hamilton College, Clinton, NY 13323, USA
[3]Department of Mechanical Engineering, University of Thessaly, 38334 Volos, Greece
[4]Computational Science and Engineering Laboratory, ETH Zürich CH-8092, Switzerland

(iD) AM, 0000-0003-0491-7329

Patient-specific modelling of haemodynamics in arterial networks has so far relied on parameter estimation for inexpensive or small-scale models. We describe here a Bayesian uncertainty quantification framework which makes two major advances: an efficient parallel implementation, allowing parameter estimation for more complex forward models, and a system for practical model selection, allowing evidence-based comparison between distinct physical models. We demonstrate the proposed methodology by generating simulated noisy flow velocity data from a branching arterial tree model in which a structural defect is introduced at an unknown location; our approach is shown to accurately locate the abnormality and estimate its physical properties even in the presence of significant observational and systemic error. As the method readily admits real data, it shows great potential in patient-specific parameter fitting for haemodynamical flow models.

## 1. Introduction

Mathematical models for haemodynamics trace back to the work of Euler, who described a one-dimensional treatment of blood flow through an arterial network with rigid tubes [1,2]; more sophisticated one-dimensional models are still used to study a variety of physio-pathological phenomena [3–10]. Computational advances have also allowed for the development of computationally intensive three-dimensional models [11–16], which have been used to accurately simulate specific human arteries (e.g. the carotid arteries [17]) and model their material properties (e.g. of cerebral arterial walls [18]). There also exist multi-component models [19], which are amenable to applications such as modelling oxygen transport to solid tumours [20] and surgical tissue flaps [21,22].

**Figure 1.** Schematic of one-dimensional artery segment.

Despite the sophistication of these approaches, there remain a number of challenges in the creation of patient-specific models using individual medical data. In particular, computational expense usually limits arterial parameter estimation to the one-dimensional class of models [2,12], which have nonetheless proven sufficiently robust to study fluid–structure interactions and viscoelasticity [7,9] and create a patient-specific model for vascular bypass surgery [23]. Several approaches exist for parameter estimation and uncertainty quantification for these models. Gradient descent has been used to estimate arterial compliance parameters [24], recovering single parameters assumed constant in space and time. Sensitivity analysis has also been used, successfully quantifying output sensitivity to various uncertainties in a stochastic flow network [25]. More recently, computational methods based upon Bayesian optimization and multi-fidelity information fusion for model inversion have been explored [26].

The chief contribution of this work is to introduce a Bayesian framework for uncertainty quantification in a bifurcating network of one-dimensional extensible arteries. The advantages of the approach are twofold. First, it uses transitional Markov chain Monte Carlo (TMCMC), a highly parallelizable algorithm for approximate sampling which allows practical uncertainty quantification even for large arterial networks [27–29]; our high-performance implementation Π4U will be shown to simultaneously and efficiently estimate several unknown parameters in this setting. Second, the approach can practically be used for Bayesian model selection, allowing for evidence-based comparison between models with distinct physical assumptions. The approach thus represents a significant advance in fitting patient-specific haemodynamical flow models.

Specifically, we consider a branching network of 19 arteries in which a structural flaw (e.g. an aneurysm) has been introduced at an unknown location. Sections 2 and 3 describe the one-dimensional blood flow model and the uncertainty quantification framework. In §4, we use the flawed model to simulate noisy observations of the flow velocity at fixed points in the network. We then use Bayesian model selection to probabilistically locate the defect within the network and accurately recover its structural properties, showing the approach to be effective even when parameters are corrupted with Gaussian noise. As the method readily admits clinical blood flow data, which have been shown to be measurable with non-invasive procedures [30–33], it shows great potential in diagnosing patient-specific structural issues in the circulatory system.

## 2. Nonlinear one-dimensional blood flow model

We first introduce the one-dimensional blood flow model. While such models can be derived via a scaling of the Navier–Stokes equations for viscous flow [34], we use here the geometry- and conservation-motivated approach described by Sherwin *et al.* [2] and Formaggia *et al.* [12]. In this approach, the viscous, incompressible flow is assumed to move only in the axial direction (i.e. along the one-dimensional artery), to exhibit axial symmetry, and to maintain constant internal pressure over orthogonal cross-sections. The artery is assumed to have low curvature and to be distensible in the radial direction. A schematic of the artery appears in figure 1.

The artery of constant length $\ell$ and position-dependent cross-sectional area $A(x, t)$ is filled with blood flowing at velocity $u(x, t)$ and with internal cross-sectional pressure $p(x, t)$, yielding the cross-sectional flux $Q(x, t) = A(x, t)u(x, t)$. Choosing $u$, $A$ and $p$ as the independent variables, the partial differential equation governing the incompressible flow can be derived from conservation of mass and momentum

and
$$\left.\begin{array}{l} \dfrac{\partial A}{\partial t} + \dfrac{\partial (Au)}{\partial x} = 0 \\[2mm] \dfrac{\partial u}{\partial t} + u\dfrac{\partial u}{\partial x} = -\dfrac{1}{\rho}\dfrac{\partial p}{\partial x} + K_r \dfrac{u}{\rho A} \end{array}\right\} \tag{2.1}$$

where $\rho$ is the flow density and $K_r$ is a parameter representing viscous resistance per unit length, here given by $K_r = -22\mu\pi$ in terms of the viscosity $\mu$ of blood and the chosen velocity profile (see [8,9] for more details).

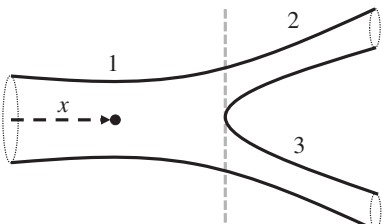

**Figure 2.** Schematic of Y-bifurcation in an arterial network.

The system is closed using a constitutive law to relate pressure and area. Using the Laplace tube law and assuming that the arterial wall is purely elastic,

$$p = p_{\text{ext}} + \frac{\sqrt{\pi}Eh}{(1 - \nu^2)\sqrt{A_0}\sqrt{A}}\left(\sqrt{A} - \sqrt{A_0}\right),$$

where $p_{\text{ext}}$ is the external pressure, $E$ is Young's modulus of the wall, $h$ is the wall thickness, $A_0$ is the relaxed cross-sectional area and $\nu$ is the Poisson ratio, here taken to be $\frac{1}{2}$. For notational simplicity, we collect the coefficient into a single stiffness parameter $B$, yielding

$$p = p_{\text{ext}} + B\left(\sqrt{A} - \sqrt{A_0}\right).$$

Equation (2.1) can then be rewritten in the form of a nonlinear hyperbolic conservation law [2]:

$$\left.\begin{array}{c}\dfrac{\partial \boldsymbol{U}}{\partial t} + \dfrac{\partial \boldsymbol{F}}{\partial x} = \boldsymbol{S}(\boldsymbol{U}), \quad \text{where} \quad \boldsymbol{U} = \begin{bmatrix} A \\ u \end{bmatrix} \\[2mm] \boldsymbol{F} = \begin{bmatrix} Au \\ \frac{u^2}{2} + \frac{p^e}{\rho} \end{bmatrix} + \begin{bmatrix} 0 \\ \frac{p^v}{\rho} \end{bmatrix}, \quad \boldsymbol{S}(\boldsymbol{U}) = \begin{bmatrix} 0 \\ K_r \frac{u}{\rho A} \end{bmatrix}, \end{array}\right\} \tag{2.2}$$

and

in terms of the elastic component $p^e(x, t)$ and viscoelastic component $p^v(x, t)$ of the pressure.

The hyperbolic system is approximated numerically using a discontinuous Galerkin method. The one-dimensional domain $\Omega = (a, b)$ is discretized into $N$ non-overlapping elements $\Omega_i = (x_i^L, x_i^R)$ such that $x_i^R = x_{i+1}^L$ and $\bigcup_{i=1}^N \bar{\Omega}_i = \bar{\Omega}$; discrete approximations to the corresponding weak formulation are found in terms of orthonormal Legendre polynomials of degree $p$ [35,36] (see, e.g. [35,37] for the advantages of this approach). Inlet and outlet boundary elements use upwind flux, while a second-order Adams–Bashforth scheme [35] is used for time integration.

To extend the model to a branching arterial network, multiple arteries are joined via coupled boundary conditions at bifurcations. An example of such a bifurcation appears in figure 2. Boundary conditions are physically motivated; mass should be conserved through bifurcations, while momentum should be continuous at the boundary, i.e.

$$A_1 u_1 = A_2 u_2 + A_3 u_3,$$

$$p_1 + \frac{1}{2}\rho u_1^2 = p_2 + \frac{1}{2}\rho u_2^2,$$

$$p_1 + \frac{1}{2}\rho u_1^2 = p_3 + \frac{1}{2}\rho u_3^2,$$

where $p_i$, $A_i$ and $u_i$ correspond to the $i$th artery. Other branching configurations appear in [2].

## 3. Bayesian uncertainty quantification

The primary practical goal of this paper is to identify structural defects in an arterial network using observations of the blood flow velocity. By varying the material properties of the arteries, perturbations to the flow can be computed via the blood flow model described in §2; in this sense, the goal is to solve the inverse problem of determining structural parameters given velocity data as model output. In real applications, these velocity data may be corrupted by noise (e.g. measurement error). Furthermore, model parameters may be fit to measurements which are themselves noisy. In this section, we introduce our recent Bayesian framework for uncertainty quantification which is amenable to the former issue (observational noise) and will prove robust to the latter (systemic noise). Section 3.3 describes the TMCMC method which forms the core of this approach: its parallelizability allows for feasible application to more expensive models, such as the model of §2, via the use of high-performance computing.

## 3.1. Parameter estimation

Denote as $M$ the mathematical model of interest, which deterministically maps a set of $n$ parameters $\underline{\theta} \in \mathbb{R}^n$ to $m$ outputs $\underline{g}(\underline{\theta}|M) \in \mathbb{R}^m$ (here, $\underline{g}$ denotes the forward problem, and so $\underline{g}(\cdot \mid M)$ is a solution to the forward problem using the model $M$). The inverse problem is then to estimate the parameters $\underline{\theta}$ given the model outputs. We assume that these model outputs have been corrupted by noise (due to e.g. measurement, computational or modelling error) as

$$\underline{D} = \underline{g}(\underline{\theta}|M) + \underline{e} \tag{3.1}$$

in terms of a random predictive error $\underline{e}$. Under the Bayesian formulation of this problem, the parameters $\underline{\theta}$ are assigned a prior distribution $\pi(\underline{\theta} \mid M)$ given any *a priori* knowledge of the parameters based on e.g. physical constraints; the posterior $p(\underline{\theta} \mid \underline{D}, M)$ that observed data $\underline{D}$ were generated by parameters $\underline{\theta}$ can then be found as

$$p(\underline{\theta} \mid \underline{D}, M) = \frac{p(\underline{D} \mid \underline{\theta}, M)\pi(\underline{\theta}|M)}{\rho(\underline{D}|M)}, \tag{3.2}$$

using the likelihood $p(\underline{D} \mid \underline{\theta}, M)$, calculated by evaluating $\underline{g}(\underline{\theta} \mid M)$ and using the form of $\underline{e}$, and the evidence $\rho(\underline{D} \mid M)$ of the model class, computed via the multidimensional integral

$$\rho(\underline{D}|M) = \int p(\underline{D} \mid \underline{\theta}, M)\pi(\underline{\theta}|M)\,\mathrm{d}\underline{\theta}.$$

In order to calculate the likelihood $p(\underline{D} \mid \underline{\theta}, M)$, we make the simplifying assumption that $\underline{e}$ is normally distributed with zero mean and covariance matrix $\Sigma$, which may itself include additional unknown parameters. Since the model outputs $\underline{g}$ are deterministic, it follows that $\underline{D}$ is also normally distributed, and so the explicit likelihood $p(\underline{D} \mid \underline{\theta}, M)$ is given by

$$p(\underline{D} \mid \underline{\theta}, M) = \frac{|\Sigma(\underline{\theta})|^{-1/2}}{(2\pi)^{m/2}}\exp\left[-\frac{1}{2}J(\underline{\theta}, \underline{D}|M)\right],$$

where

$$J(\underline{\theta}, \underline{D}|M) = [\underline{D} - \underline{g}(\underline{\theta}|M)]^T \Sigma^{-1}(\underline{\theta})[\underline{D} - \underline{g}(\underline{\theta}|M)]$$

is the weighted measure of fit between the model predictions and the measured data, $|\cdot|$ denotes determinant, and the parameter set $\underline{\theta}$ is augmented to include parameters that are involved in the structure of the covariance matrix $\Sigma$.

## 3.2. Model selection

The Bayesian approach to uncertainty quantification is especially useful in the context of model selection. The evidence $\rho(\underline{D} \mid M)$ which appears in equation (3.2) is a measure of the degree to which the model $M$ can explain the data $\underline{D}$; when $M$ is one particular model in a parametrized class $\mathcal{M}$ of models, the evidence can be used to derive a distribution on models. Let $Pr(M_i)$ be a prior distribution on models in the class $\mathcal{M}$. The posterior $\Pr(M_i \mid \underline{D})$ can again be derived from Bayes' theorem:

$$\Pr(M_i | \underline{D}) = \frac{\rho(\underline{D}|M_i)\Pr(M_i)}{p(\underline{D}|\mathcal{M})},$$

where $p(\underline{D}|M) = \sum_i \rho(\underline{D}|M_i)Pr(M_i)$ is a normalization constant. Intuitively, $\Pr(M_i \mid \underline{D})$ is a distribution which describes the probability of the data $\underline{D}$ having been generated from model $M_i$ (as opposed to another model $M_j$) under the assumption that at least one model in $\mathcal{M}$ is the true model, i.e. was actually used to generate the data. If a uniform prior is assumed on models, this posterior is directly proportional to the evidence $\rho(\underline{D} \mid M_i)$, and so model selection is 'free' when the evidence is already calculated for parameter estimation [29,38–40].

## 3.3. Transitional Markov chain Monte Carlo

While there exist many approaches to solving the proposed Bayesian inverse problem (e.g. [41–43]), few are constrained by the main computational barrier in this application: the complex forward problem $\underline{g}$ (here, the blood flow model of §2) which appears in the fitness $J(\underline{\theta}, \underline{D} \mid M)$. The TMCMC algorithm,

**Algorithm 1** TMCMC

1: **procedure** TMCMC [27,29]

2: BEGIN, SET $j = 0$, $q_0 = 0$

3: **Generate** $\{\underline{\theta}_{0,k}, k = 1, \dots, N_0\}$ from prior $f_0(\underline{\theta}) = \pi(\underline{\theta}|M)$ and compute likelihood $p(\underline{D}|\underline{\theta}_{0,k}, M)$ for each sample.

4: *loop*:

5: **WHILE** $q_{j+1} \leq 1$ **DO:**

6:     **Analyze** samples $\{\underline{\theta}_{j,k}, k = 1, \dots, N_j\}$ to determine $q_{j+1}$, weights $\overline{w}(\underline{\theta}_{j,k})$, covariance $\Sigma_j$, and estimator $S_j$ of $\mathbb{E}[w(\underline{\theta}_{j,k})]$.

7:     **Resample** based on samples available in stage $j$ using the plausibility weights and the Metropolis algorithm in order to generate samples for stage $j + 1$ and compute likelihood $p(\underline{D}|\underline{\theta}_{j+1,k}, M)$ for each.

8:     **if** $q_{j+1} > 1$ **then**

9:         BREAK,

10:     **else**

11:         $j = j + 1$

12:         **goto** *loop*.

13:     **end**

14: **END**

developed by Ching & Chen [27], is a useful approach in this context; by smoothly transitioning to the target distribution (the posterior $p(\underline{\theta} \mid \underline{D}, M)$) from the prior $\pi(\underline{\theta} \mid M)$, repeated evaluations of the forward problem $\underline{g}$ in regions of low probability are avoided. Our implementation, $\Pi$4U, further takes advantage of the parallelizability of TMCMC with a highly efficient architecture for task sharing (appendix A).

To accomplish a smooth transition, we define a series of intermediate distributions:

$$f_j(\underline{\theta}) \sim [p(\underline{D} \mid \underline{\theta}, M)]^{q_j} \cdot \pi(\underline{\theta} \mid M), \quad j = 0, \dots, \lambda$$
$$0 = q_0 < q_1 < \cdots < q_\lambda = 1.$$

The original TMCMC algorithm is summarized above in algorithm 1. It begins by taking $N_0$ samples $\underline{\theta}_{0,k}$ from the prior distribution $f_0(\underline{\theta}) = \pi(\underline{\theta} \mid M)$. For each stage $j$ of the algorithm, the current samples are used to compute the plausibility weights $w(\underline{\theta}_{j,k})$ as

$$w(\underline{\theta}_{j,k}) = \frac{f_{j+1}(\underline{\theta}_{j,k})}{f_j(\underline{\theta}_{j,k})} = [p(\underline{D} \mid \underline{\theta}_{j,k}, M)]^{q_{j+1} - q_j}.$$

Recent literature suggests that $q_{j+1}$, which determines how smoothly the intermediate distributions transition to the posterior, should be taken to make the covariance of the plausibility weights at stage $j$ smaller than a tolerance covariance value, often 1.0 [29].

Next, the algorithm calculates the average $S_j$ of the plausibility weights, the normalized plausibility weights $\overline{w}(\underline{\theta}_{j,k})$ and the scaled covariance $\Sigma_j$ of the samples $\underline{\theta}_{j,k}$, which is used to produce the next generation of samples $\underline{\theta}_{j+1,k}$:

$$S_j = \frac{1}{N_j} \sum_{k=1}^{N_j} w(\underline{\theta}_{j,k}),$$

$$\overline{w}(\underline{\theta}_{j,k}) = \frac{w(\underline{\theta}_{j,k})}{\sum_{k=1}^{N_j} w(\underline{\theta}_{j,k})} = \frac{w(\underline{\theta}_{j,k})}{(N_j S_j)}$$

and

$$\Sigma_j = b^2 \sum_{k=1}^{N_j} \overline{w}(\underline{\theta}_{j,k})[\underline{\theta}_{j,k} - \underline{\mu}_j][\underline{\theta}_{j,k} - \underline{\mu}_j]^{\mathrm{T}}.$$

$\Sigma_j$ is calculated using the sample mean $\underline{\mu}_j$ and a scaling factor $b$, usually 0.2 [29].

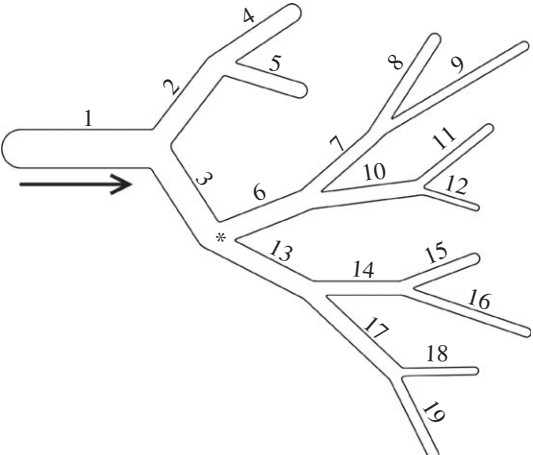

**Figure 3.** Schematic of arterial network (not to scale). The 19 arteries have varied lengths (ranging from 0.026 to 0.17 m) and cross-sectional areas (ranging from $10^{-5}$ to $10^{-6}$ m$^2$). The star shows an example measurement location at a bifurcation.

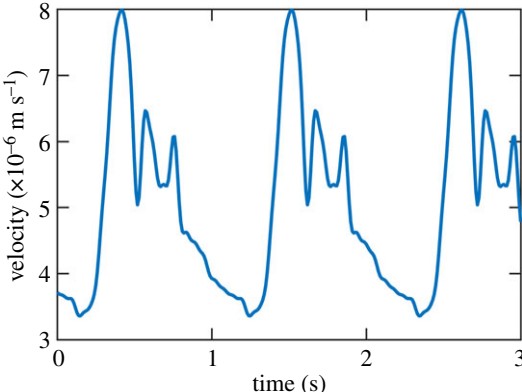

**Figure 4.** Inflow boundary condition for blood velocity (m s$^{-1}$) corresponding to three cardiac cycles.

The algorithm then generates $N_{j+1}$ samples $\hat{\underline{\theta}}_{j+1,k}$ by randomly selecting from the previous generation $\{\underline{\theta}_{j,k}\}$ such that $\hat{\underline{\theta}}_{j+1,\ell} = \underline{\theta}_{j,k}$ with probability $\overline{w}(\underline{\theta}_{j,k})$. These samples are selected independently at random, so any parameter can be selected multiple times—call $n_{j+1,k}$ the number of times $\underline{\theta}_{j,k}$ is selected. Each unique sample is used as the starting point of an independent Markov chain of length $n_{j+1,k}$ generated using the Metropolis algorithm with target distribution $f_j$ and a Gaussian proposal distribution with covariance $\Sigma_j$ centred at the current value.

Finally, the samples $\underline{\theta}_{j+1,k}$ are generated for the Markov chains, with $n_{j+1,k}$ samples drawn from the chain starting at $\underline{\theta}_{j,k}$, yielding $N_{j+1}$ total samples. The algorithm then either moves forward to generation $j+1$ or terminates if $q_{j+1} > 1$.

## 4. Results

We now apply the Bayesian framework of §3 to the blood flow model of §2. In particular, we study the example 19-artery network shown in figure 3. The solution for our deterministic model, given by (2.2) and solved using a discontinuous Galerkin method with time step $\Delta t_1 = 0.00004$ s, plays the role of $\underline{g}$ in the model prediction equation (3.1). Measurements of the flow velocity are taken at $N$ specified locations which vary by experiment and occur with a sampling period of $\Delta t_2 = 1600\Delta t_1 = 0.064$ s.

Blood (viscosity $\mu^* = 0.0045$, with asterisks denoting reference values) in the network begins with zero velocity and is driven by a specified inflow velocity at the beginning of the first artery: a sum of trigonometric polynomials, shown in figure 4, which approximates the flow for three cardiac cycles [9]. The length of three cycles (approx. 3.3 s) allows for a total of 52 velocity data per measurement location using the sampling period $\Delta t_2$, and so the output space of $\underline{g}$ has dimension 52$N$. The outflow condition is a fully absorbing boundary condition, described in more detail in [8].

We consider three free parameters: the blood viscosity $\mu$, the arterial stiffness parameter $B$ and the relaxed cross-sectional area $A_0$ (the last two of which vary by artery). A structural defect (e.g. an aneurysm or stenosis) can be modelled by varying the stiffness or relaxed area of a particular artery; to better emphasize the degree to which a flawed artery has been modified by its defect, results will use the scaled stiffness $\beta = B/B^*$ (with respect to the reference stiffness $B^*$) and scaled cross-sectional area $\alpha = A_0/A_0^*$ (with respect to the reference area $A_0^*$), and so arteries with no defect will have $\beta = \alpha = 1$.

We use our implementation of Bayesian uncertainty quantification to examine a number of questions in the context of this forward model, focusing in particular on the ability of uncertainty quantification to identify the location of structural flaws within the network using only noisy measurements of the flow velocity. To test the effectiveness of our implementation in these experiments thus requires noisy data $\underline{D}$ corresponding to a known truth; we use here synthetic data generated from the same model but with known, fixed parameters. It should be stressed that the approach is easily modified to admit real data and that there exist multiple practical methods for measuring blood flow velocities from *in vivo* arteries [30–33]. Section 4.3 will show that flaws can be located accurately even when the parameters used to generate the synthetic data are significantly perturbed from the parameters used to perform uncertainty quantification.

Explicitly, observed data $\underline{D}$ are generated as

$$D_k = v_k + \sigma \boldsymbol{\epsilon}_k, \tag{4.1}$$

where $D_k$ is the noisy observation at time $t_k$, $v_k$ is the flow velocity at time $t_k$, $\epsilon_k$ is a zero-mean, unit-variance Gaussian random variable and $\sigma$ is the noise level. Here, we choose $\sigma$ to be a fraction $\sigma = 0.01\eta$ (or sometimes $0.05\eta$) of the standard deviation $\eta$ of all velocity data $v_k$.

In the following results, we use our implementation of uncertainty quantification to generate 500 samples from the posterior distribution $p(\theta \mid D, M)$ in a variety of scenarios. Posterior distributions are used for parameter estimation (§4.1) and to identify structural flaws via Bayesian model selection (§§4.2 and 4.3). Recovered posterior means, denoted with a hat (e.g. $\hat{\beta}$), are used as parameter estimates in our analysis.

## 4.1. Parameter estimation

We first consider a basic case of parameter estimation to illustrate the feasibility of the approach. Specifically, we estimate the blood viscosity $\mu$ and the scaled stiffness $\beta_2$ of artery 2 (see figure 3 for artery labels) assuming all other parameters are fixed to their reference values. The noisy data used, corrupted according to (4.1) with noise level $\sigma = 0.01\eta$, are sampled from a single location at the start of artery 6. As described in §3, we choose $\sigma$ as an additional free parameter, requiring the approach to recover the noise level in addition to the target model parameters. A uniform distribution on $[0.5, 1.5] \times [0.5, 1.5] \times [0, 1]$ in the parameter space $(\mu, \beta_2, \sigma)$ is used as the parameter prior $\pi$.

To determine the effect of the choice of sampling location, we additionally consider separate cases using data obtained from the start of arteries 1 and 8; for notational clarity, we refer to as $O_i$ the case of observing the upflow end of artery $i$.

The results for the case $O_6$ appear in figure 5. $\mu$ and $\beta_2$ are positively correlated in the posterior, i.e. simultaneously raising or lowering both the blood viscosity and the stiffness of artery 2 yields qualitatively similar observed data. Intuitively, in order to maintain a consistent rate of flow, a viscous flow necessitates more rigid artery walls.

Numerical results for $O_1$, $O_6$ and $O_8$ are summarized in table 1. The recovered posterior means of $(\mu, \beta_2, \sigma)$ were (0.00410, 0.972, 0.00074), (0.00449, 0.997, 0.00074) and (0.00447, 0.994, 0.00077), respectively, closely matching the true values $\mu^* = 0.0045$ and $\beta_2^* = 1.0$ ($\sigma^*$ differed by experiment due to differences in flow velocity by location: 0.00074, 0.00074 and 0.00077 for $O_1$, $O_6$ and $O_8$, respectively). To quantify the degree of uncertainty in each parameter's posterior distribution, we compute a coefficient of variation, defined as the ratio of the single-parameter posterior's standard deviation to its mean (denoting the results $u_{\hat{\mu}}$, $u_{\hat{\beta}_2}$ and $u_{\hat{\sigma}}$); here, $O_6$ and $O_8$ recover parameters with comparatively lower uncertainty than $O_1$, whose measurements were largely dominated by the inflow boundary condition. Nonetheless, in all cases, the reference values used to generate the synthetic data were within one standard deviation of the recovered posterior means.

## 4.2. Locating structural flaws with model selection

Given the practicality of parameter estimation and its intermediate estimation of the model evidence $\rho(\underline{D} \mid M)$, the Bayesian model selection framework described in §3.2 is a natural approach to locating structural flaws in the arterial network. Namely, define as $M_i$ the model in which the scaled stiffness $\beta$ of artery $i$ has

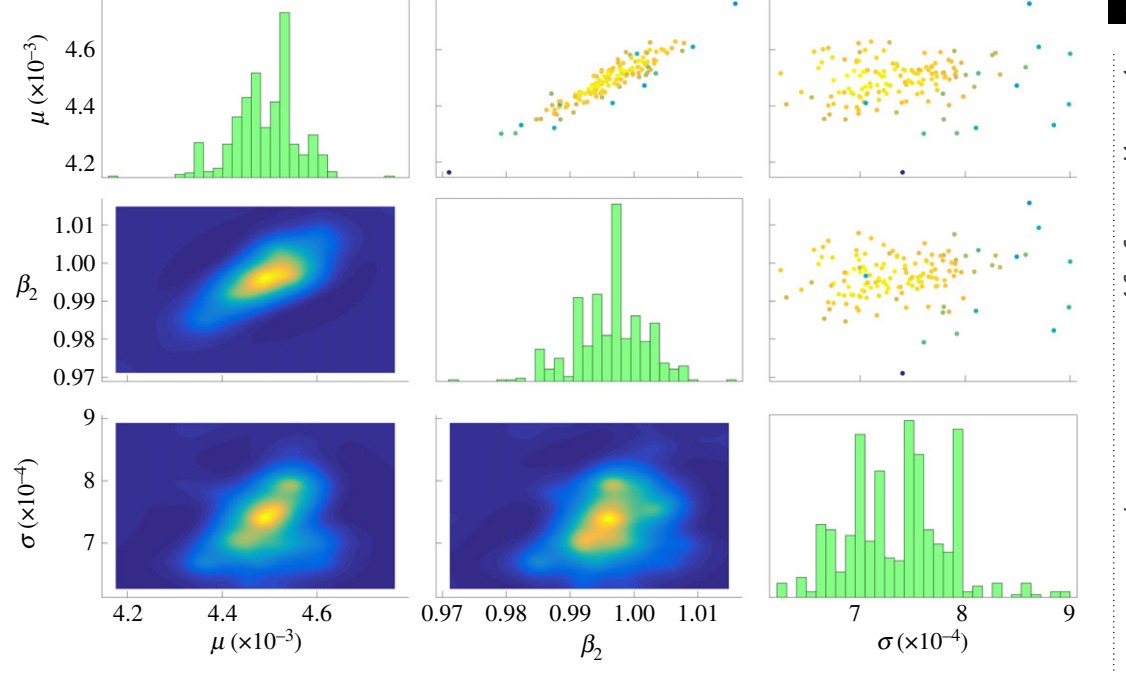

**Figure 5.** Parameter estimation results for blood viscosity $\mu$, arterial stiffness $\beta_2$ for artery 2, and noise level $\sigma$ using corrupted reference data from the beginning of artery 6 ($O_6$). Figures on the diagonal show histograms for each parameter. Subfigures below the diagonal show the marginal joint densities for each pair of parameters, while subfigures above the diagonal show the samples used in the final (convergent) stage of TMCMC. Colours correspond to likelihoods, with yellow likely and blue unlikely. For visual intuition on TMCMC convergence, see appendix B.

**Table 1.** Posterior means and uncertainties for parameter estimation on the 19-artery network for three cases $O_i$ (noise level $\sigma = 0.01\eta$).

| data observations | $\hat{\mu}$ | $u_{\hat{\mu}}$ (%) | $\hat{\beta}_2$ | $u_{\hat{\beta}_2}$ (%) | $\hat{\sigma}$ | $u_{\hat{\sigma}}$ (%) |
|---|---|---|---|---|---|---|
| $O_1$ | 0.004102 | 11.86 | 0.97175 | 3.48 | 0.00074218 | 5.87 |
| $O_6$ | 0.004493 | 1.41 | 0.99663 | 0.51 | 0.00073577 | 6.30 |
| $O_8$ | 0.004470 | 2.05 | 0.99367 | 0.70 | 0.00076524 | 7.65 |

been perturbed from its reference value by an unknown amount, corresponding to e.g. an aneurysm or stenosis. Parameter estimation as in §4.1 can be used to recover the perturbed stiffness which best matches the observed data, simultaneously yielding an estimate for the evidence $\rho(\underline{D} \mid M_i)$ of model $M_i$. Letting $\mathcal{M}$ be the collection of $M_i$ for various arteries $i$ in the network, the model selection distribution $\Pr(M_i \mid \underline{D})$ is a probabilistic measure of the likelihood of the structural defect occurring in artery $i$ (as opposed to a different artery $j$). The class $\mathcal{M}$ can easily be augmented with additional models; here, we also consider a model $M_{i:j}$ which freely varies the stiffness of two arteries $i$ and $j$.

We first consider generating data $\underline{D}$ from a reference model using $\beta_6 = 0.5$ and $\beta_i = 1$, $i \neq 6$, i.e. a model in which the scaled stiffness of artery 6 has been halved from its reference value. We consider three cases for data collection: a two-sensor configuration using data from the end of arteries 1 and 7, a three-sensor configuration using data from the end of arteries 1, 7 and 13, and a four-sensor configuration using velocity data from the end of arteries 1, 7, 10 and 13. In each case, no sampling locations are adjacent to the damaged artery. The noise level is again chosen as $\sigma = 0.01\eta$, i.e. 1% Gaussian noise, and we employ the same uniform prior on $[0, 3] \times [0, 1]$ for the parameters $(\beta, \sigma)$.

Table 2 presents numerical results for six flaw models $M_3$, $M_6$, $M_7$, $M_{11}$, $M_{13}$ and $M_{6:7}$ when taking flow measurements from the ends of arteries 1 and 7. Models $M_6$ and $M_{6:7}$, both of which include the correct defect location in artery 6, are assigned the largest probabilities under the model selection

**Table 2.** Numerical results for identification of a $\beta_6 = 0.5$ aneurysm using noisy data from the ends of arteries 1 and 7 (noise level $\sigma = 0.01\eta$).

| prediction model | $\hat{\beta}$ | $u_{\hat{\beta}}$ (%) | $\hat{\sigma}$ | $u_{\hat{\sigma}}$ (%) | log evidence | $\Pr(M_j \mid \underline{D})$ |
|---|---|---|---|---|---|---|
| $M_3$ | 0.741 | 2.82 | 0.000786 | 6.31 | 583.2 | 0.00009 |
| $M_6$ | 0.499 | 6.61 | 0.000729 | 3.73 | 592.5 | 0.99949 |
| $M_7$ | 1.055 | 3.14 | 0.000923 | 4.27 | 567.7 | $\sim 0$ |
| $M_{11}$ | 1.022 | 0.76 | 0.000892 | 5.96 | 566.3 | $\sim 0$ |
| $M_{13}$ | 0.587 | 5.81 | 0.000867 | 5.04 | 578.3 | $\sim 0$ |
| $M_{6:7}$ | [0.501, 1.036] | [3.75, 2.15] | 0.000719 | 5.16 | 584.7 | 0.00042 |

**Table 3.** Numerical results for identification of a $\beta_6 = 0.5$ aneurysm using data from the ends of arteries 1, 7 and 13 (noise level $\sigma = 0.01\eta$).

| prediction model | $\hat{\beta}$ | $u_{\hat{\beta}}$ (%) | $\hat{\sigma}$ | $u_{\hat{\sigma}}$ (%) | log evidence | $\Pr(M_j \mid \underline{D})$ |
|---|---|---|---|---|---|---|
| $M_3$ | 0.722 | 2.29 | 0.000811 | 4.81 | 875.6 | 0.00001 |
| $M_6$ | 0.501 | 4.31 | 0.000770 | 5.23 | 886.3 | 0.70756 |
| $M_7$ | 1.056 | 2.19 | 0.000956 | 4.64 | 853.1 | $\sim 0$ |
| $M_{11}$ | 1.005 | 0.60 | 0.000949 | 5.02 | 848.8 | $\sim 0$ |
| $M_{13}$ | 0.901 | 1.91 | 0.000887 | 5.06 | 858.3 | $\sim 0$ |
| $M_{6:7}$ | [0.505, 1.031] | [3.45, 2.33] | 0.000770 | 3.58 | 885.5 | 0.29242 |

posterior ($\Pr(M_j \mid \underline{D}) = 0.99949$, 0.00042, respectively); recovered parameter estimates $\hat{\beta}_6 = 0.499$ ($M_6$) and $\hat{\beta}_6 = 0.501$ ($M_{6:7}$) for the wall stiffness of the damaged artery were accurate to within one standard deviation. Though $M_{6:7}$ assumes a second defect in artery 7, the posterior mean estimated the stiffness to be similar to the reference value ($\hat{\beta}_7 = 1.036$). $M_3$, $M_7$, $M_{11}$ and $M_{13}$ are not able to accurately match the observed data, and so require a significantly higher noise level $\sigma$ to explain differences between the evaluated and observed velocities. For this reason, these models are assigned negligible mass by $\Pr(M_i \mid \underline{D})$.

Note that the model $M_{6:7}$ contains $M_6$ in the sense that it can predict any combination of parameter values which $M_6$ can predict. In this light, the relatively higher evidence for $M_6$ over the broader error model $M_{6:7}$ is in keeping with theoretical results available for Bayesian model class selection wherein over-parametrized model classes are penalized due to Occam's factor [38].

Table 3 illustrates the corresponding results for a three-sensor configuration using blood flow velocity data from the ends of arteries 1, 7 and 13. The additional data collected from artery 13 significantly reduce the (already small) probabilities assigned to models other than $M_6$ and $M_{6:7}$. Interestingly, leveraging information from the end of artery 13, which is in a parallel tree (rather than directly upstream or downstream) from the damaged artery, has the effect of shifting mass from $M_6$ to $M_{6:7}$ in the model posterior, finding $\Pr(M_6 \mid \underline{D}) = 0.708$ and $\Pr(M_{6:7} \mid \underline{D}) = 0.292$. Nonetheless, both $M_6$ and $M_{6:7}$ estimate the damaged stiffness $\beta_6$ accurately (0.501 and 0.505, respectively), with $M_{6:7}$ again finding the stiffness $\beta_7$ of the undamaged artery 7 to be largely unchanged (1.031).

Finally, table 4 shows numerical results when velocity data are sampled at four monitoring locations: at the ends of arteries 1, 7, 10 and 13. The additional data from the end of artery 10 drive the model probabilities assigned to $M_3$, $M_7$, $M_{11}$ and $M_{13}$ down further ($< 10^{-8}$), rendering them orders of magnitude smaller than the probabilities assigned to $M_6$ and $M_{6:7}$ (0.9996 and 0.0004, respectively). The estimated scaled stiffness remains accurate to within one standard deviation, with $M_6$ and $M_{6:7}$ finding $\hat{\beta}_6 = 0.520$ and 0.521, respectively, and $M_{6:7}$ again estimates the stiffness of artery 7 to be only slightly perturbed ($\hat{\beta}_7 = 1.027$).

Taken together, these configurations support two conclusions about Bayesian model selection for flaw identification: first, that increasing the number of locations at which data are sampled reduces the probabilities assigned to incorrect models, and second, that model selection can accurately

**Table 4.** Numerical results for identification of a $\beta_6 = 0.5$ aneurysm using data from the ends of arteries 1, 7, 10 and 13 (noise level $\sigma = 0.01\eta$).

| prediction model | $\hat{\beta}$ | $u_{\hat{\beta}}$ (%) | $\hat{\sigma}$ | $u_{\hat{\sigma}}$ (%) | log evidence | $Pr(M_j \mid \underline{D})$ |
|---|---|---|---|---|---|---|
| $M_3$ | 0.760 | 2.60 | 0.000817 | 3.93 | 1169.9 | $\sim 0$ |
| $M_6$ | 0.520 | 2.57 | 0.000762 | 3.48 | 1187.3 | 0.9996 |
| $M_7$ | 1.046 | 2.72 | 0.000926 | 3.86 | 1144.5 | $\sim 0$ |
| $M_{11}$ | 0.996 | 0.19 | 0.000932 | 3.58 | 1134.1 | $\sim 0$ |
| $M_{13}$ | 0.890 | 1.83 | 0.000902 | 3.47 | 1148.8 | $\sim 0$ |
| $M_{6:7}$ | [0.521, 1.027] | [3.98, 2.49] | 0.000771 | 3.63 | 1179.5 | 0.0004 |

**Table 5.** Numerical results for area-based identification of an $\alpha_6 = 1.5$ aneurysm using data from the ends of arteries 1, 7, 10 and 13 (noise level $\sigma = 0.01\eta$).

| prediction model | $\hat{\alpha}$ | $u_{\hat{\alpha}}$ (%) | $\hat{\sigma}$ | $u_{\hat{\sigma}}$ (%) | log evidence | $Pr(M_j \mid \underline{D})$ |
|---|---|---|---|---|---|---|
| $M_3$ | 1.126 | 26.29 | 0.00554 | 3.40 | 781.0 | $\sim 0$ |
| $M_6$ | 1.508 | 0.45 | 0.00078 | 4.33 | 1183.6 | 0.998 |
| $M_7$ | 0.976 | 0.17 | 0.00415 | 4.15 | 827.9 | $\sim 0$ |
| $M_{11}$ | 1.048 | 0.48 | 0.00471 | 3.41 | 810.0 | $\sim 0$ |
| $M_{13}$ | 1.014 | 0.17 | 0.00518 | 3.18 | 793.0 | $\sim 0$ |
| $M_{6:7}$ | [1.507, 1.000] | [0.52, 0.031] | 0.00077 | 3.24 | 1177.6 | 0.002 |

determine the defect location and magnitude for a variety of sensor configurations, including configurations which do not sample from at or near the defect location.

### 4.2.1. Model selection for cross-sectional area

As previously suggested, aneurysms and stenoses can also be modelled by adjusting the initial cross-sectional area of an artery rather than its stiffness. Ideally, the Bayesian framework for model selection should provide similar results when the stiffnesses $\beta$ are fixed and models $M_i$ instead allow the scaled cross-sectional area $\alpha$ of the defective artery to be perturbed. In what follows, we examine similar scenarios to the above in the case where, rather than reducing its wall stiffness, the relaxed cross-sectional area of artery 6 is altered. A uniform prior on $[0, 3] \times [0, 1]$ is used for the parameters $(\alpha, \sigma)$.

We first consider the case $\alpha_6 = 1.5$, i.e. an aneurysm in which the defective artery (again, artery 6 in the reference model) has become enlarged by 50%. Noisy flow velocity data are collected from the ends of arteries 1, 7, 10 and 13, as in the final case of the previous section; results appear in table 5. $M_6$ and $M_{6:7}$ are again the most likely models ($Pr(M_j \mid \underline{D}) = 0.998$ and 0.002, respectively), suggesting that the previous results do not rely on the specific choice of the parameter $\beta$. Other models were assigned negligible probabilities. Similarly to the results for reduced stiffness, both $M_6$ and $M_{6:7}$ accurately recover the defect magnitude ($\hat{\alpha}_6 = 1.508, 1.507$, respectively), and $M_{6:7}$ finds artery 7 to be unchanged ($\hat{\alpha}_7 = 1.000$).

We then consider the same scenario for a reduction $\alpha_6 = 0.5$ in the cross-sectional area of artery 6, i.e. a stenosis in which the defective artery has narrowed by 50%. Results are summarized in table 6. $M_6$ and $M_{6:7}$ recover the reduced area accurately ($\hat{\alpha}_6 = 0.500, 0.501$, respectively) and are assigned the highest model evidence ($Pr(M_j \mid \underline{D}) \approx 1.00$ and $\sim 10^{-4}$, respectively).

Table 7 shows results for the same magnitude stenosis ($\alpha_6 = 0.5$) with increased observational noise level $\sigma = 0.05\eta$. The log evidence of models $M_6$ and $M_{6:7}$ is sharply reduced compared to table 6, though $M_6$ and $M_{6:7}$ remain the most probable models under the model selection posterior, with $Pr(M_j \mid \underline{D}) = 0.983$ and 0.017, respectively. Both models additionally recover the reduced area accurately ($\hat{\alpha}_6 = 0.502$ and 0.503, respectively) despite the increased noise.

Finally, table 8 considers the case of a smaller-magnitude stenosis ($\alpha_6 = 0.8$). Results were similar to those of table 7, with accurate recovery of location ($Pr(M_j \mid \underline{D}) = 0.99999$ and 0.00001 for $M_6$, $M_{6:7}$,

**Table 6.** Numerical results for area-based identification of an $\alpha_6 = 0.5$ stenosis using data from the ends of arteries 1, 7, 10 and 13 (noise level $\sigma = 0.01\eta$).

| prediction model | $\hat{\alpha}$ | $u_{\hat{\alpha}}$ (%) | $\hat{\sigma}$ | $u_{\hat{\sigma}}$ (%) | log evidence | $\Pr(M_j \mid \underline{D})$ |
|---|---|---|---|---|---|---|
| $M_3$ | 0.072 | 8.00 | 0.023 | 4.04 | 478.9 | ~0 |
| $M_6$ | 0.500 | 0.05 | 0.00076 | 3.16 | 1178.0 | 1.00 |
| $M_7$ | 1.127 | 0.78 | 0.0195 | 4.83 | 513.3 | ~0 |
| $M_{11}$ | 0.790 | 2.37 | 0.0215 | 3.72 | 493.8 | ~0 |
| $M_{13}$ | 0.941 | 0.73 | 0.0238 | 3.33 | 475.2 | ~0 |
| $M_{6:7}$ | [0.501, 1.000] | [0.08, 0.03] | 0.00076 | 3.30 | 1164.6 | ~0 |

**Table 7.** Numerical results for area-based identification of an $\alpha_6 = 0.5$ stenosis using data from the ends of arteries 1, 7, 10 and 13 (noise level $\sigma = 0.05\eta$).

| prediction model | $\hat{\alpha}$ | $u_{\hat{\alpha}}$ (%) | $\hat{\sigma}$ | $u_{\hat{\sigma}}$ (%) | log evidence | $\Pr(M_j \mid \underline{D})$ |
|---|---|---|---|---|---|---|
| $M_3$ | 0.070 | 4.98 | 0.0234 | 2.56 | 481.1 | ~0 |
| $M_6$ | 0.502 | 0.37 | 0.00376 | 4.10 | 849.6 | 0.983 |
| $M_7$ | 1.118 | 0.90 | 0.0199 | 2.78 | 512.5 | ~0 |
| $M_{11}$ | 0.800 | 2.09 | 0.0221 | 4.33 | 492.5 | ~0 |
| $M_{13}$ | 0.940 | 0.88 | 0.0234 | 3.97 | 474.8 | ~0 |
| $M_{6:7}$ | [0.503, 1.000] | [0.40, 0.23] | 0.00381 | 4.69 | 845.6 | 0.017 |

**Table 8.** Numerical results for area-based identification of an $\alpha_6 = 0.8$ stenosis using data from the ends of arteries 1, 7, 10 and 13 (noise level $\sigma = 0.01\eta$).

| prediction model | $\hat{\alpha}$ | $u_{\hat{\alpha}}$ (%) | $\hat{\sigma}$ | $u_{\hat{\sigma}}$ (%) | log evidence | $\Pr(M_j \mid \underline{D})$ |
|---|---|---|---|---|---|---|
| $M_3$ | 0.175 | 4.13 | 0.00493 | 3.71 | 809.0 | ~0 |
| $M_6$ | 0.801 | 0.18 | 0.00075 | 4.68 | 1185.1 | 0.99999 |
| $M_7$ | 1.024 | 0.15 | 0.00404 | 3.78 | 838.0 | ~0 |
| $M_{11}$ | 0.955 | 0.45 | 0.00447 | 3.92 | 820.5 | ~0 |
| $M_{13}$ | 0.987 | 0.17 | 0.00488 | 3.89 | 802.7 | ~0 |
| $M_{6:7}$ | [0.802, 1.000] | [0.16, 0.033] | 0.00077 | 3.11 | 1173.7 | 0.00001 |

respectively) and magnitude ($\hat{\alpha}_6 = 0.801$ and $0.802$). As in the previous area-modification scenarios, $M_{6:7}$ found artery 7 to be unaffected ($\hat{\alpha}_7 = 1.000$), thereby coinciding with the single defect model $M_6$.

## 4.3. Locating defects with misspecified models

Results have so far assumed the model selection framework is provided the reference values for all model parameters, i.e. the non-defective stiffness and area of each artery are known. In a scenario using real-world data, these 'known' values must themselves be estimated from noisy measurements. A final but crucial test of the robustness of the framework is thus to perform experiments in which the reference parameters used by the method are incorrect, and so no combination of free parameters is capable of reproducing the observed data.

**Table 9.** Numerical results for area-based identification of an $\alpha_6 = 1.5$ aneurysm using data from the ends of arteries 1, 7, 10 and 13 (noise level $\sigma = 0.01\eta$) with misspecified cross-sectional areas (perturbed with noise level $\sigma_\alpha = 0.01$).

| prediction model | $\hat{\alpha}$ | $u_{\hat{\alpha}}$ (%) | $\hat{\sigma}$ | $u_{\hat{\sigma}}$ (%) | log evidence | $\Pr(M_j \mid \underline{D})$ |
|---|---|---|---|---|---|---|
| $M_3$ | 1.312 | 29.2 | 0.00761 | 3.39 | 711.6 | $\sim 0$ |
| $M_6$ | 2.047 | 0.89 | 0.00217 | 3.98 | 972.9 | 0.907 |
| $M_7$ | 0.970 | 0.23 | 0.00628 | 3.48 | 748.7 | $\sim 0$ |
| $M_{11}$ | 1.075 | 0.56 | 0.00645 | 3.76 | 746.9 | $\sim 0$ |
| $M_{13}$ | 1.016 | 0.29 | 0.00725 | 3.53 | 716.8 | $\sim 0$ |
| $M_{6:7}$ | [2.123, 1.002] | [3.27, 0.11] | 0.00216 | 4.32 | 970.6 | 0.093 |

**Table 10.** Numerical results for area-based identification of an $\alpha_6 = 0.5$ stenosis using data from the ends of arteries 1, 7, 10 and 13 (noise level $\sigma = 0.01\eta$) with misspecified cross-sectional areas (perturbed with noise level $\sigma_\alpha = 0.01$).

| prediction model | $\hat{\alpha}$ | $u_{\hat{\alpha}}$ (%) | $\hat{\sigma}$ | $u_{\hat{\sigma}}$ (%) | log evidence | $\Pr(M_j \mid \underline{D})$ |
|---|---|---|---|---|---|---|
| $M_3$ | 0.816 | 7.29 | 0.0220 | 4.36 | 485.7 | $\sim 0$ |
| $M_6$ | 0.515 | 0.17 | 0.0021 | 3.22 | 973.6 | 0.999 |
| $M_7$ | 1.118 | 0.66 | 0.0181 | 3.44 | 529.0 | $\sim 0$ |
| $M_{11}$ | 0.808 | 1.72 | 0.0207 | 3.41 | 505.7 | $\sim 0$ |
| $M_{13}$ | 0.943 | 0.69 | 0.0221 | 3.14 | 490.5 | $\sim 0$ |
| $M_{6:7}$ | [0.517, 1.003] | [0.23, 0.001] | 0.0021 | 3.87 | 966.5 | 0.001 |

We now revisit the cases of §4.2.1, beginning with the case of an $\alpha_6 = 1.5$ aneurysm in artery 6. In addition to corrupting observed flow velocities with additive Gaussian noise, we now additionally corrupt the parameters themselves: the initial cross-sectional area $\alpha_k$ for each artery $k$ is noised as

$$\alpha_k = \alpha_k^*(1 + \sigma_\alpha \epsilon_k), \tag{4.2}$$

where $\alpha_k^*$ is the reference value, $\epsilon_k$ is again a standard normal random variable, and $\sigma_\alpha$ is the parameter noise level. The structural parameters used to generate the synthetic data ($\alpha_k$ from equation (4.2)) thus differ from the fixed values used in the defect models $M_i$ ($\alpha_k^*$).

As before, Bayesian model selection is performed assuming the prediction equation (3.1), which is now misspecified (it assumes correctness of the reference parameters $\alpha_i^*$). As a result, the $\hat{\sigma}$ estimated by posterior samples must now capture the effects of both the true observational noise level $\sigma$ and the parameter noise level $\sigma_\alpha$.

Table 9 shows numerical results for Bayesian model selection in this setting. Despite the misspecification, $M_6$ and $M_{6:7}$ again dominate the model posterior, with $\Pr(M_6 \mid \underline{D}) = 0.907$ and $\Pr(M_{6:7} \mid \underline{D}) = 0.093$, respectively. Both overestimate the defect magnitude ($\hat{\alpha}_6 = 2.047, 2.123$, respectively), though $M_{6:7}$ again estimates artery 7 to be unaffected ($\hat{\alpha}_7 = 1.002$). We note that some error in $\hat{\alpha}_6$ is expected, as it attempts to fit observations from the noised-parameter model and thus varies significantly depending on the particular values of $\alpha_k$ from equation (4.2). Despite this effect, identification of the location appears robust to perturbation of model parameters, with all other models assigned negligible probability ($\Pr(M_i \mid \underline{D}) \sim 0$).

Turning to the second case ($\alpha_6 = 0.5$), model selection again successfully locates the defect despite the misspecification (table 10), with $M_6$ assigned nearly all mass by the model selection posterior. In this case, parameter estimation recovers the defect magnitude accurately ($\hat{\alpha}_6 = 0.515$). In keeping with previous results, defect model $M_{6:7}$ finds a similar reduction in cross-sectional area for the damaged artery ($\hat{\alpha}_6 = 0.517$) and little change in the defect-free artery ($\hat{\alpha}_7 = 1.003$).

The third case repeated the $\alpha_6 = 0.5$ experiment with increased observational noise $\sigma = 0.05\eta$; results for the same case with parameter noise (now also increased to $\sigma_\alpha = 0.05$) are shown in table 11. $M_{6:7}$ is significantly more likely than in previous cases ($\Pr(M_{6:7} \mid \underline{D}) = 0.783$), though $M_6$ is still assigned all remaining posterior mass ($\Pr(M_6 \mid \underline{D}) = 0.217$). The recovered uncertainties $u_{\hat{\alpha}}$ are significantly higher

**Table 11.** Numerical results for area-based identification of an $\alpha_6 = 0.5$ stenosis using data from the ends of arteries 1, 7, 10 and 13 (noise level $\sigma = 0.05\eta$) with misspecified model parameters (perturbed with noise level $\sigma_\alpha = 0.05$).

| prediction model | $\hat{\alpha}$ | $u_{\hat{\alpha}}$ (%) | $\hat{\sigma}$ | $u_{\hat{\sigma}}$ (%) | log evidence | $\Pr(M_j \mid \underline{D})$ |
|---|---|---|---|---|---|---|
| $M_3$ | 1.668 | 45.9 | 0.0195 | 4.35 | 518.5 | ~0 |
| $M_6$ | 0.597 | 0.80 | 0.0111 | 3.31 | 636.6 | 0.217 |
| $M_7$ | 1.085 | 0.62 | 0.00076 | 4.82 | 558.5 | ~0 |
| $M_{11}$ | 0.894 | 1.52 | 0.0185 | 3.82 | 527.1 | ~0 |
| $M_{13}$ | 0.947 | 0.51 | 0.0175 | 3.62 | 534.9 | ~0 |
| $M_{6:7}$ | [0.609, 1.015] | [1.22, 0.44] | 0.0108 | 2.22 | 637.9 | 0.783 |

**Table 12.** Numerical results for area-based identification of an $\alpha_6 = 0.8$ stenosis using data from the ends of arteries 1, 7, 10 and 13 (noise level $\sigma = 0.01\eta$) with misspecified model parameters (perturbed with noise level $\sigma_\alpha = 0.01$).

| prediction model | $\hat{\alpha}$ | $u_{\hat{\alpha}}$ (%) | $\hat{\sigma}$ | $u_{\hat{\sigma}}$ (%) | log evidence | $\Pr(M_j \mid \underline{D})$ |
|---|---|---|---|---|---|---|
| $M_3$ | 1.767 | 22.2 | 0.00388 | 3.57 | 852.4 | ~0 |
| $M_6$ | 0.865 | 0.35 | 0.00204 | 3.36 | 974.9 | 0.969 |
| $M_7$ | 1.017 | 0.13 | 0.00320 | 3.87 | 889.5 | ~0 |
| $M_{11}$ | 0.979 | 0.24 | 0.00369 | 3.61 | 860.3 | ~0 |
| $M_{13}$ | 0.988 | 0.12 | 0.00345 | 3.49 | 869.5 | ~0 |
| $M_{6:7}$ | [0.869, 1.003] | [0.50, 0.076] | 0.00209 | 3.22 | 971.5 | 0.031 |

than in table 10 due to the higher level of noise, with both $M_6$ and $M_{6:7}$ underestimating the magnitude of the damage ($\hat{\alpha}_6 = 0.597, 0.609$, respectively).

Finally, table 12 shows results for the fourth case ($\alpha_6 = 0.8$) in the presence of $\sigma_\alpha = 0.01$ parameter noise. While model selection again recovers the correct defect location ($\Pr(M_j \mid \underline{D}) = 0.969, 0.031$ for $M_6$, $M_{6:7}$, respectively), the smaller-magnitude stenosis proves more challenging for parameter estimation, with $\hat{\alpha}_6 = 0.865$ and $0.869$, respectively, notably underestimating the magnitude of the damage.

# 5. Discussion

Taken together, the results describe a robust approach for uncertainty quantification in the context of arterial networks. The model selection posterior universally assigned the highest probabilities (by several orders of magnitude) only to those models which included the true defect location, even in cases where simulated data were sparse, noisy and poorly located. The Bayesian uncertainty quantification framework thus appears a powerful tool for comparing and fitting models.

Though all results were generated using simulated noisy data, they simultaneously suggest that the approach would prove useful for real-world inference. The experiments outlined in §4.2 show the method to successfully recover parameter values (often within one standard deviation) and identify the defect location in a range of sampling cases which varied sensor numbers and locations, and so the approach is not reliant on a particular set of observed data which may not be realistically attainable. Results were also consistent when using alternative magnitudes and parametrizations of arterial defects (the scaled cross-sectional area $\alpha$ and boundary stiffness $\beta$) and using models which considered different numbers of defects (in particular, the two-defect model $M_{6:7}$ which consistently found the 'defective' artery 7 to be largely unaltered). We note that the approach readily facilitates the incorporation of real data, which can be used in place of simulated data without otherwise altering the method. A natural extension of this work is therefore direct application to medical datasets.

It is worth emphasizing the role of model selection in our approach. The general Bayesian inverse problem, which simultaneously considers structural properties of all arterial segments, cannot be feasibly

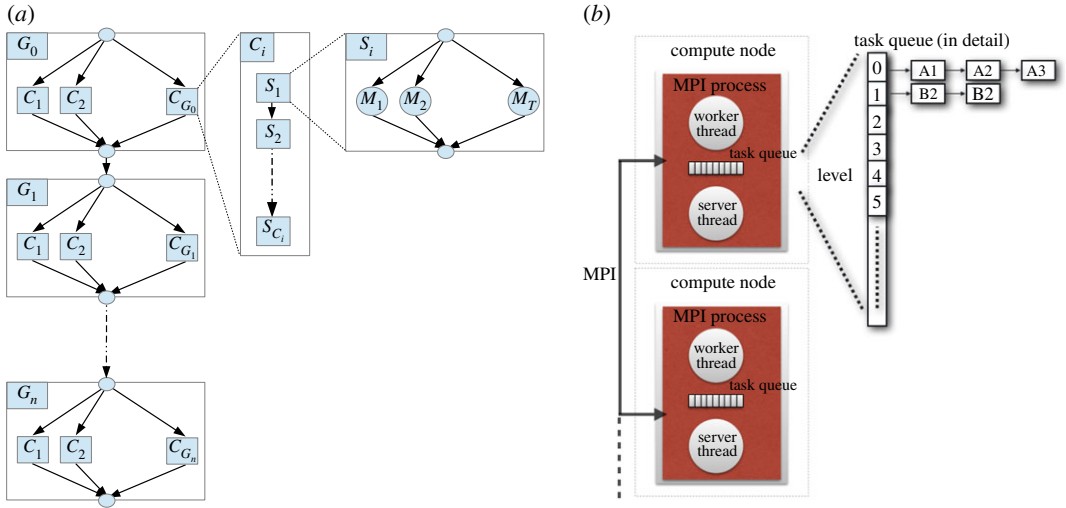

**Figure 6.** Task graph of the TMCMC algorithm (*a*) and parallel architecture of the TORC library (*b*).

solved due to its dimensionality. In §4.2, we instead propose a set of simple models which fix most model parameters to 'known' values, then select among these models to locate flaws; while this approach has a massive computational advantage, it also makes the strong assumption that the properties of healthy artery segments are known exactly. We investigate robustness of the approach to this assumption in §4.3, which found inference to remain effective even when 'known' model parameters were assumed incorrectly.

While model selection has here served primarily as a computational tool, it is more commonly used to select between distinct physical models of a system given noisy data. Robustness to this form of model misspecification, i.e. the assumption of a physically incorrect model, is a more complex issue begetting a range of additional techniques such as discrepancy terms [44] and posterior predictive assessment [45]. Future work should investigate this more general sense of robustness using alternative flow models, network structures, and defects in more localized arterial subdomains.

Data accessibility. All data needed to evaluate the conclusions in the paper are present in the paper. The code for the Bayesian Uncertainty Quantification and Optimization framework *Π*4U is available online at: https://github.com/cselab/pi4u.

Authors' contributions. K.L. and C.B. developed code, ran simulations, visualized and analysed data and drafted the manuscript; C.P., P.K. and A.M. edited the manuscript and conceived, designed and coordinated the study. All authors gave final approval for publication.

Competing interests. The authors declare no competing interests.

Funding. K.L. and A.M. were partially supported by the NSF through grant nos. DMS-1521266 and DMS-1552903. C.P. was supported by the European Union's Horizon 2020 research and innovation programme under the Marie Sklodowska-Curie grant agreement no. 764547. P.K. was supported by the European Research Council Advanced Investigator Award (grant no. 34117).

Acknowledgements. Parts of this research were conducted using computational resources and services at the Center for Computation and Visualization, Brown University.

# Appendix A. High-performance implementations

*Π*4U [29,46] is a platform-agnostic task-based UQ framework that supports nested parallelism and automatic load balancing in large-scale computing architectures. The software is open-source and includes HPC implementations for both multi-core and GPU clusters of algorithms such as TMCMC and approximate Bayesian computational subset simulation. The irregular, dynamic and multi-level task-based parallelism of the algorithms (figure 6a) is expressed and fully exploited by means of the TORC run-time library [47]. TORC is a software library for programming and running unaltered task-parallel programs on both shared and distributed memory platforms. TORC orchestrates the scheduling of function evaluations on the cluster nodes (figure 6b). The parallel framework includes multiple features, most prominently the inherent load balancing, fault tolerance and high reusability. The TMCMC method within *Π*4U is able to achieve an overall parallel efficiency of more than 90% on 1024 compute nodes of Swiss supercomputer

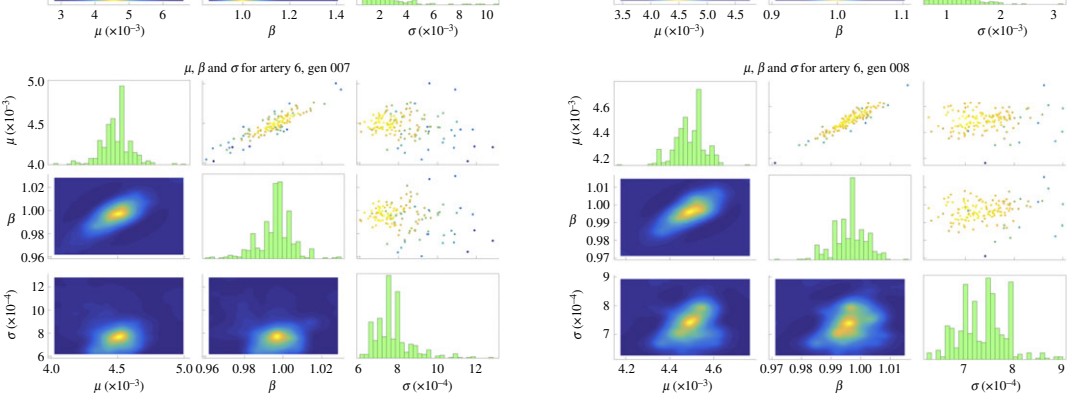

**Figure 7.** Samples from eight generations of TMCMC in §4.1. Plot limits are not constant (the sampled region of high probability monotonically decreases in size).

Piz Daint running hybrid MPI+GPU molecular simulation codes with highly variable time-to-solution between simulations with different interaction parameters.

# Appendix B. Convergence of transitional Markov chain Monte Carlo

Figure 7 shows eight generations of TMCMC convergence from parameter estimation in §4.1. Samples are initialized from the prior, i.e. sampled uniformly from all feasible parameter values. The transitional target distribution yields gradual coalescence on regions of high likelihood, eventually resampling directly from the posterior.

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
