## [Reviewer comments · Royal Society Open Science]

Review History

RSOS-182229.R0 (Original submission)

Review form: Reviewer 1

Is the manuscript scientifically sound in its present form?

Yes

Are the interpretations and conclusions justified by the results?

Yes

Is the language acceptable?

Yes

Is it clear how to access all supporting data?

No

Do you have any ethical concerns with this paper?

No

Have you any concerns about statistical analyses in this paper?

No

Recommendation?

Accept with minor revision (please list in comments)

Comments to the Author(s)

The paper is very well written and brings a very important contribution bridging the gap between modelling of the arterial network and Bayesian model selection.

Section 3.3 on TMCMC does not reference the literature appropriately and it is very hard to understand who introduced what, and if the method used in your current work actually has any improvements or changes over the standard paper by Ching & Chen.

In particular the following issues should be addressed:

- Does the algorithm you use adjust the sample weights after each MCMC step in order to reduce the average bias in the model evidence estimate?
- It is not clear if there is any burn-in period applied to the MCMC step.
- Algorithm 1 states the TMCMC algorithm, without any mentioning of MCMC steps or burn-in periods.

However the following line appears in the next page of the paper: "Each unique sample is used as the starting point of an independent Markov chain", and there is no mention of MCMC steps in the algorithmic description.

- A discussion needs to be added on how the TMCMC algorithm relates to the algorithms called "parallel tempering" and "particle filtering". They are extremely related to it and share many features. Why TMCMC and not thermodynamic integration to estimate the model evidence? How do the computational complexities differ?

The results section is very well structured and explained.

It would be interesting to also see some trace-plots of the MCMC iterations in the Appendix.

The results are convincing, but I would like to also see a discussion on what would happen if, rather than considering some fixed models identified by given constrained parametrizations of the arterial model from Section 2, and then selecting between them, one just used the full arterial model.

In other words: given data D from a patient with an aneurysm, it would be interesting to estimate all of the model parameters (for the full arterial network model) using TMCMC and see if the estimated values identify the aneurysm in the right location, without recurring to model selection.

This would be a way to solve the problem of not having the right model in the class of models to be compared and missing a potential clinical diagnosis, with clear health risks for the patient. We should keep in mind that we want to reduce the probability of False negatives!

Of course the starting arterial model could be completely wrong itself, because of the structural assumptions, but in order to account for the mismatch between the model and the data there is a literature on calibration of computer models and learning the "discrepancy term". Perhaps a reference on this could be useful, as very likely the assumptions of the model will not hold for real data.

[citation]

Kennedy, M., & O'Hagan, A. (2001). Bayesian Calibration of Computer Models. *Journal of the Royal Statistical Society. Series B (Statistical Methodology)*, 63(3), 425-464. Retrieved from <http://www.jstor.org/stable/2680584>

Reproducibility and code of the results section:

The code needs to be available on GitHub / Bitbucket / Gitlab, for reproducibility, along with the simulation settings and corresponding random number generator seeds.

Appendix:

In the appendix, a link to the Pi4U repository would be helpful for the reader.

[reference needed]

Ching, J., Chen, Y.-C., 2007. Transitional Markov chain Monte Carlo method for Bayesian model updating, model class selection, and model averaging. *Journal of Engineering Mechanics* 133 (7), 816-832.

[if any of their adjustments are used?]

Betz W., Papaioannou I., Straub D. (2016): Transitional Markov chain Monte Carlo: Observations and improvements. *Journal of Engineering Mechanics, ASCE*, 142(5): 04016016.

Review form: Reviewer 2

Is the manuscript scientifically sound in its present form?

Yes

Are the interpretations and conclusions justified by the results?

Yes

Is the language acceptable?

Yes

Is it clear how to access all supporting data?

No

Do you have any ethical concerns with this paper?

No

Have you any concerns about statistical analyses in this paper?

Yes

Recommendation?

Accept with minor revision (please list in comments)

Comments to the Author(s)

Summary

Larson et al. present a study on detecting arterial wall abnormalities using Bayesian inference. They demonstrate their results on synthetic data case studies. They use an efficient parallel sampling scheme to solve the inverse problem, and the model is a partial differential equation. They also consider the performance of their methods under some forms of model misspecification.

Comments.

Overall impression.

I enjoyed reading this manuscript and think it is a useful contribution to the literature. It reads well and has a clear rationale. The methods are justified and appropriate for the more specific claims made.

My only real concerns relate to some of the more general claims about the novelty and robustness of the approach. These are discussed below.

Major comments

Framework

The authors state (p. 27) that they introduce 'our recent Bayesian framework for uncertainty quantification which is amenable to [dealing with noise and model misspecification].

As far as I can tell, the basic framework itself is standard Bayesian inference for inverse problems, and is not in itself 'novel', while the sampling method is more novel. Perhaps this is what the authors mean by 'framework' (e.g. the parallel computing aspect)?

Regardless, this is absolutely fine, but I would emphasise that the basic Bayesian framework is standard while the particular computational aspect is more 'novel' (rather than just saying 'our framework') and add some citations to standard references on Bayesian inverse problems at this point e.g.

- Stuart, A. M. (2010). Inverse problems: a Bayesian perspective. *Acta numerica*, 19, 451-559.

- Kaipio, J., & Somersalo, E. (2006). *Statistical and computational inverse problems* (Vol. 160). Springer Science & Business Media.

- Tarantola, A. (2005). *Inverse problem theory and methods for model parameter estimation* (Vol. 89). SIAM.

or other preferred/standard references.

Robustness to misspecification

Regarding the 'robustness to misspecification' aspect, I would be more tentative. As shown in e.g. the Kaipio and Somersalo book mentioned above and related work, model misspecification can have a large effect on Bayesian inference. This is obviously the case for other inverse problems/statistical inference methods too.

In the Kaipio and Somersalo reference and related work the authors explicitly construct approximation error models to account for/guard against this misspecification. There are a number of other articles along these lines, i.e. trying to explicitly deal with misspecification, that the authors do not appear to engage with. This usually means some form of 'wrong' model is used, more severe than just a parameter being slightly the wrong value but the correct equations used.

As far as I can tell, in contrast to the above, in the present work the authors simply 'ignore' misspecification and hope for the best, i.e. they do not explicitly account for misspecification during the inference procedures or framework. Which again, is often fine! Depending on how sensitive a particular quantity of interest is with respect to errors in other 'nuisance' factors. In this present work this is checked via synthetic computational case studies, which again is fine, but does not provide especially clear guidelines for when this does and does not work - there should certainly be cases where misspecification really does matter, e.g. a completely qualitatively different model is used. The authors do give one case where the 'misspecification' matters a bit. I would like to see more explicit and extreme examples where the misspecification really *does* matter, and more guidelines for, or discussion of, when this might happen (and why).

The main implication of this is that *I would recommend replacing some of the more general claims of robustness and accuracy of the model selection process by more cautious discussions in terms of 'sensitivity studies'* (of particular quantities with respect to errors in some particular quantities).

Finally, as indicated above, model misspecification more commonly refers to when the model structure itself is misspecified rather than simply setting one or more of the other parameters in the correct equations to (slightly) 'wrong' values. Have the authors considered this case? E.g. are there competing models, or models with the wrong branching structure etc which could be used to represent more systematic misspecification? Some discussion or example in this direction would greatly add to the paper.

Alternatively, I think the article framing should shift the emphasis of the robustness aspect to be about (relatively limited) 'sensitivity' studies rather than more general claims concerning 'robustness' to (true) 'misspecification'.

Robustness more generally

I would also be cautious about general claims to robustness and the reliability of model selection procedure used, i.e. the use of model evidence/Bayes factors.

For example, there are aspects of Bayesian inference that can be very susceptible to misspecification, see e.g.

- Owhadi, H., Scovel, C., & Sullivan, T. (2015). On the brittleness of Bayesian inference. *SIAM Review*, 57(4), 566-582.

and the issue of the robustness of Bayesian model selection is widely discussed in the statistical literature and quite controversial, see e.g.

- Kass, R. E., & Raftery, A. E. (1995). Bayes factors. *Journal of the American statistical association*, 90(430), 773-795.

- Rubin, D. B. (1995). Avoiding model selection in Bayesian social research. *Sociological methodology*, 25, 165-173.

- Gelman, A., Carlin, J. B., Stern, H. S., Dunson, D. B., Vehtari, A., & Rubin, D. B. (2013). *Bayesian data analysis*. Chapman and Hall/CRC. (Chapter 7 in particular)

- Gelman, A., & Shalizi, C. R. (2013). Philosophy and the practice of Bayesian statistics. *British Journal of Mathematical and Statistical Psychology*, 66(1), 8-38.

One issue in particular is that the *model* Bayes factor/marginal likelihoods/evidence can be especially sensitive to the *parameter* prior specification, even when the parameter posterior is not. It is usually recommended that (at least) sensitivity to this is checked.

Thus I would like to see the authors do this, i.e. *check the sensitivity of their model selection procedure to the parameter priors used to marginalise out the parameters*.

Again, this is not to say that the present work is not justified, but that I would prefer a more cautious discussion in terms of the particular sensitivity studies carried out in this work, and a few more carried out. There will almost certainly be cases where the framework is not robust, and it is important to a) convey this and b) outline/indicate under what sort of situations this might be expected to happen.

Data space (and graphical) checks

As part of the sensitivity studies carried out here, I would quite like to see prior or posterior *predictive* checks (i.e. data space checks of the implied predictive distributions over observable state variables).

I would also prefer these (and the parameter space distributions) to be graphical rather than tabular where possible. Graphical displays to me usually give a far better indication of the manner(s) in which the model is performing well or poorly.

E.g. in one context a 'log evidence' of 781 vs 1183 might be great, but in a different context the same numbers could be driven by some undesirable aspects of the model (see above about sensitivity to parameter prior specification). I would be concerned about real-world data and model misspecification driving the evidence values, without graphical and data space checks.

Some good references on predictive checks and graphical display of these are:

- Gelman, A., Meng, X. L., & Stern, H. (1996). Posterior predictive assessment of model fitness via realized discrepancies. *Statistica sinica*, 733-760.

- Gelman, A., Carlin, J. B., Stern, H. S., Dunson, D. B., Vehtari, A., & Rubin, D. B. (2013). *Bayesian data analysis*. Chapman and Hall/CRC.

- Gelman, A. (2004). Exploratory data analysis for complex models. *Journal of Computational and Graphical Statistics*, 13(4), 755-779.

Minor comments

- Line 31/31. Say 'Equation (1)' rather than starting the sentence with (1).
- I would probably prefer to use e.g. V instead of J in the equations in lines 7-13, as I would usually reserve J for the Jacobian.
- In section 3.3. the authors should reference their PI4U article(s). Some discussion of/references to/comparison to other sampling methods used in large-scale Bayesian inverse problems would also be useful.
- Data availability. As far as I can see the authors code and data are not available in a manner such that other people could run and reproduce their results. I think the authors should make enough code/data available (e.g. via GitHub or similar) such that someone could easily reproduce at least some portion of this analysis (while perhaps being limited by available parallel computing facilities).

Decision letter (RSOS-182229.R0)

23-Aug-2019

Dear Dr Matzavinos

On behalf of the Editors, I am pleased to inform you that your Manuscript RSOS-182229 entitled "Detection of arterial wall abnormalities via Bayesian model selection" has been accepted for publication in Royal Society Open Science subject to minor revision in accordance with the referee suggestions. Please find the referees' comments at the end of this email.

The reviewers and handling editors have recommended publication, but also suggest some minor revisions to your manuscript. Therefore, I invite you to respond to the comments and revise your manuscript.

- Ethics statement

- Data accessibility

If you wish to submit your supporting data or code to Dryad (<http://datadryad.org/>), or modify your current submission to dryad, please use the following link:
<http://datadryad.org/submit?journalID=RSOS&manu=RSOS-182229>

- **Competing interests**

- **Authors' contributions**

- **Acknowledgements**

- **Funding statement**

Because the schedule for publication is very tight, it is a condition of publication that you submit the revised version of your manuscript before 01-Sep-2019. Please note that the revision deadline will expire at 00.00am on this date. If you do not think you will be able to meet this date please let me know immediately.

When submitting your revised manuscript, you will be able to respond to the comments made by the referees and upload a file "Response to Referees" in "Section 6 - File Upload". You can use this to document any changes you make to the original manuscript. In order to expedite the

processing of the revised manuscript, please be as specific as possible in your response to the referees. We strongly recommend uploading two versions of your revised manuscript:

Kind regards,
Alice Power
Editorial Coordinator
Royal Society Open Science

on behalf of Dr Xiaoyu Luo (Associate Editor) and Mark Chaplain (Subject Editor)
openscience@royalsociety.org

Associate Editor Comments to Author (Dr Xiaoyu Luo):

Please make minor revisions as requested by the reviewers.

Reviewer comments to Author:

Reviewer: 1

The paper is very well written and brings a very important contribution bridging the gap between modelling of the arterial network and Bayesian model selection.

Section 3.3 on TMCMC does not reference the literature appropriately and it is very hard to understand who introduced what, and if the method used in your current work actually has any improvements or changes over the standard paper by Ching & Chen.

In particular the following issues should be addressed:

- Does the algorithm you use adjust the sample weights after each MCMC step in order to reduce the average bias in the model evidence estimate?
- It is not clear if there is any burn-in period applied to the MCMC step.
- Algorithm 1 states the TMCMC algorithm, without any mentioning of MCMC steps or burn-in periods.

However the following line appears in the next page of the paper: "Each unique sample is used as the starting point of an independent Markov chain", and there is no mention of MCMC steps in the algorithmic description.

- A discussion needs to be added on how the TMCMC algorithm relates to the algorithms called "parallel tempering" and "particle filtering". They are extremely related to it and share many features. Why TMCMC and not thermodynamic integration to estimate the model evidence? How do the computational complexities differ?

The results section is very well structured and explained.

It would be interesting to also see some trace-plots of the MCMC iterations in the Appendix. The results are convincing, but I would like to also see a discussion on what would happen if, rather than considering some fixed models identified by given constrained parametrizations of the arterial model from Section 2, and then selecting between them, one just used the full arterial model.

In other words: given data D from a patient with an aneurysm, it would be interesting to estimate all of the model parameters (for the full arterial network model) using TMCMC and see if the estimated values identify the aneurysm in the right location, without recurring to model selection.

This would be a way to solve the problem of not having the right model in the class of models to be compared and missing a potential clinical diagnosis, with clear health risks for the patient. We should keep in mind that we want to reduce the probability of False negatives!

Of course the starting arterial model could be completely wrong itself, because of the structural assumptions, but in order to account for the mismatch between the model and the data there is a literature on calibration of computer models and learning the "discrepancy term". Perhaps a

reference on this could be useful, as very likely the assumptions of the model will not hold for real data.

[citation]

Kennedy, M., & O'Hagan, A. (2001). Bayesian Calibration of Computer Models. *Journal of the Royal Statistical Society. Series B (Statistical Methodology)*, 63(3), 425-464. Retrieved from <http://www.jstor.org/stable/2680584>

Reproducibility and code of the results section:

The code needs to be available on GitHub / Bitbucket / Gitlab, for reproducibility, along with the simulation settings and corresponding random number generator seeds.

Appendix:

In the appendix, a link to the Pi4U repository would be helpful for the reader.

[reference needed]

Ching, J., Chen, Y.-C., 2007. Transitional Markov chain Monte Carlo method for Bayesian model updating, model class selection, and model averaging. *Journal of Engineering Mechanics* 133 (7), 816-832.

[if any of their adjustments are used?]

Betz W., Papaioannou I., Straub D. (2016): Transitional Markov chain Monte Carlo: Observations and improvements. *Journal of Engineering Mechanics, ASCE*, 142(5): 04016016.

Reviewer: 2

Comments to the Author(s)

Summary

Larson et al. present a study on detecting arterial wall abnormalities using Bayesian inference. They demonstrate their results on synthetic data case studies. They use an efficient parallel sampling scheme to solve the inverse problem, and the model is a partial differential equation. They also consider the performance of their methods under some forms of model misspecification.

Comments.

Overall impression.

I enjoyed reading this manuscript and think it is a useful contribution to the literature. It reads well and has a clear rationale. The methods are justified and appropriate for the more specific claims made.

My only real concerns relate to some of the more general claims about the novelty and robustness of the approach. These are discussed below.

Major comments

Framework

The authors state (p. 27) that they introduce 'our recent Bayesian framework for uncertainty quantification which is amenable to [dealing with noise and model misspecification].

As far as I can tell, the basic framework itself is standard Bayesian inference for inverse problems,

and is not in itself 'novel', while the sampling method is more novel. Perhaps this is what the authors mean by 'framework' (e.g. the parallel computing aspect)?

Regardless, this is absolutely fine, but I would emphasise that the basic Bayesian framework is standard while the particular computational aspect is more 'novel' (rather than just saying 'our' framework) and add some citations to standard references on Bayesian inverse problems at this point e.g.

- Stuart, A. M. (2010). Inverse problems: a Bayesian perspective. *Acta numerica*, 19, 451-559.
- Kaipio, J., & Somersalo, E. (2006). *Statistical and computational inverse problems* (Vol. 160). Springer Science & Business Media.
- Tarantola, A. (2005). *Inverse problem theory and methods for model parameter estimation* (Vol. 89). SIAM.

or other preferred/standard references.

Robustness to misspecification

Regarding the 'robustness to misspecification' aspect, I would be more tentative. As shown in e.g. the Kaipio and Somersalo book mentioned above and related work, model misspecification can have a large effect on Bayesian inference. This is obviously the case for other inverse problems/statistical inference methods too.

In the Kaipio and Somersalo reference and related work the authors explicitly construct approximation error models to account for/guard against this misspecification. There are a number of other articles along these lines, i.e. trying to explicitly deal with misspecification, that the authors do not appear to engage with. This usually means some form of 'wrong' model is used, more severe than just a parameter being slightly the wrong value but the correct equations used.

As far as I can tell, in contrast to the above, in the present work the authors simply 'ignore' misspecification and hope for the best, i.e. they do not explicitly account for misspecification during the inference procedures or framework. Which again, is often fine! Depending on how sensitive a particular quantity of interest is with respect to errors in other 'nuisance' factors. In this present work this is checked via synthetic computational case studies, which again is fine, but does not provide especially clear guidelines for when this does and does not work - there should certainly be cases where misspecification really does matter, e.g. a completely qualitatively different model is used. The authors do give one case where the 'misspecification' matters a bit. I would like to see more explicit and extreme examples where the misspecification really *does* matter, and more guidelines for, or discussion of, when this might happen (and why).

The main implication of this is that *I would recommend replacing some of the more general claims of robustness and accuracy of the model selection process by more cautious discussions in terms of 'sensitivity studies'* (of particular quantities with respect to errors in some particular quantities).

Finally, as indicated above, model misspecification more commonly refers to when the model structure itself is misspecified rather than simply setting one or more of the other parameters in the correct equations to (slightly) 'wrong' values. Have the authors considered this case? E.g. are there competing models, or models with the wrong branching structure etc which could be used

to represent more systematic misspecification? Some discussion or example in this direction would greatly add to the paper.

Alternatively, I think the article framing should shift the emphasis of the robustness aspect to be about (relatively limited) 'sensitivity' studies rather than more general claims concerning 'robustness' to (true) 'misspecification'.

Robustness more generally

I would also be cautious about general claims to robustness and the reliability of model selection procedure used, i.e. the use of model evidence/Bayes factors.

For example, there are aspects of Bayesian inference that can be very susceptible to misspecification, see e.g.

- Owhadi, H., Scovel, C., & Sullivan, T. (2015). On the brittleness of Bayesian inference. *SIAM Review*, 57(4), 566-582.

and the issue of the robustness of Bayesian model selection is widely discussed in the statistical literature and quite controversial, see e.g.

- Kass, R. E., & Raftery, A. E. (1995). Bayes factors. *Journal of the American statistical association*, 90(430), 773-795.

- Rubin, D. B. (1995). Avoiding model selection in Bayesian social research. *Sociological methodology*, 25, 165-173.

- Gelman, A., Carlin, J. B., Stern, H. S., Dunson, D. B., Vehtari, A., & Rubin, D. B. (2013). *Bayesian data analysis*. Chapman and Hall/CRC. (Chapter 7 in particular)

- Gelman, A., & Shalizi, C. R. (2013). Philosophy and the practice of Bayesian statistics. *British Journal of Mathematical and Statistical Psychology*, 66(1), 8-38.

One issue in particular is that the *model* Bayes factor/marginal likelihoods/evidence can be especially sensitive to the *parameter* prior specification, even when the parameter posterior is not. It is usually recommended that (at least) sensitivity to this is checked.

Thus I would like to see the authors do this, i.e. *check the sensitivity of their model selection procedure to the parameter priors used to marginalise out the parameters*.

Again, this is not to say that the present work is not justified, but that I would prefer a more cautious discussion in terms of the particular sensitivity studies carried out in this work, and a few more carried out. There will almost certainly be cases where the framework is not robust, and it is important to a) convey this and b) outline/indicate under what sort of situations this might be expected to happen.

Data space (and graphical) checks

As part of the sensitivity studies carried out here, I would quite like to see prior or posterior *predictive* checks (i.e. data space checks of the implied predictive distributions over observable state variables).

I would also prefer these (and the parameter space distributions) to be graphical rather than

tabular where possible. Graphical displays to me usually give a far better indication of the manner(s) in which the model is performing well or poorly.

E.g. in one context a 'log evidence' of 781 vs 1183 might be great, but in a different context the same numbers could be driven by some undesirable aspects of the model (see above about sensitivity to parameter prior specification). I would be concerned about real-world data and model misspecification driving the evidence values, without graphical and data space checks.

Some good references on predictive checks and graphical display of these are:

- Gelman, A., Meng, X. L., & Stern, H. (1996). Posterior predictive assessment of model fitness via realized discrepancies. *Statistica sinica*, 733-760.

- Gelman, A., Carlin, J. B., Stern, H. S., Dunson, D. B., Vehtari, A., & Rubin, D. B. (2013). *Bayesian data analysis*. Chapman and Hall/CRC.

- Gelman, A. (2004). Exploratory data analysis for complex models. *Journal of Computational and Graphical Statistics*, 13(4), 755-779.

Minor comments

- Line 31/31. Say 'Equation (1)' rather than starting the sentence with (1).

- I would probably prefer to use e.g. V instead of J in the equations in lines 7-13, as I would usually reserve J for the Jacobian.

- In section 3.3. the authors should reference their PI4U article(s). Some discussion of/references to/comparison to other sampling methods used in large-scale Bayesian inverse problems would also be useful.

- Data availability. As far as I can see the authors code and data are not available in a manner such that other people could run and reproduce their results. I think the authors should make enough code/ data available (e.g. via GitHub or similar) such that someone could easily reproduce at least some portion of this analysis (while perhaps being limited by available parallel computing facilities).

Author's Response to Decision Letter for (RSOS-182229.R0)

See Appendix A.

Decision letter (RSOS-182229.R1)

17-Sep-2019

Dear Dr Matzavinos,

I am pleased to inform you that your manuscript entitled "Detection of arterial wall abnormalities via Bayesian model selection" is now accepted for publication in Royal Society Open Science.

on behalf of Dr Xiaoyu Luo (Associate Editor) and Mark Chaplain (Subject Editor)
openscience@royalsociety.org

Associate Editor Comments to Author (Dr Xiaoyu Luo):
Associate Editor: 1
Comments to the Author:
(There are no comments.)

Reviewer comments to Author:

Appendix A

Response to referees' comments for the paper:

“Detection of arterial wall abnormalities via Bayesian model selection”

Manuscript ID: RSOS-182229 R1

Karen Larson, Clark Bowman, Costas Papadimitriou, Petros Koumoutsakos, Anastasios Matzavinos

The authors are grateful to both reviewers for a thorough and constructive criticism. Each reviewer offered several good comments and questions; we are happy to re-submit the accompanying revised manuscript which we feel has addressed their major concerns.

Reviewer #1

Section 3.3 on TMCMC does not reference the literature appropriately and it is very hard to understand who introduced what, and if the method used in your current work actually has any improvements or changes over the standard paper by Ching and Chen.

The authors agree that Section 3.3 presently strikes an unfortunate middle-ground between methodology and implementation. TMCMC as described in the paper is the algorithm of Ching and Chen; while we did cite their work in intermediate steps, it was certainly not clear that no modifications were made to the original method. The original aspect of this paper (with respect to TMCMC) is the efficient parallel implementation $\Pi4U$ which is cited and briefly discussed in Appendix A. We have edited this section to better emphasize which aspects of the approach are original.

In particular the following issues should be addressed: 1a) Does the algorithm you use adjust the sample weights after each MCMC step in order to reduce the average bias in the model evidence estimate?

The sample weights are indeed adjusted after each step (Step 6, Algorithm 1).

1b) It is not clear if there is any burn-in period applied to the MCMC step.

A burn-in period is used to allow a chain to reach a region of high probability before sampling. In TMCMC, samples gradually adjust as the distribution transitions from prior to posterior such that samples are continuously in regions of high probability (ideally avoiding bottlenecks entirely), so no burn-in period is necessary.

1c) Algorithm 1 states the TMCMC algorithm, without any mentioning of MCMC steps or burn-in periods. However the following line appears in the next page of the paper: “Each unique sample is used as the starting point of an independent Markov chain”, and there is no mention of MCMC steps in the algorithmic description.

We thank the reviewer for pointing out this discrepancy. We have adjusted the phrasing of Step 7, Algorithm 1 to emphasize when Metropolis is used.

1d) A discussion needs to be added on how the TMCMC algorithm relates to the algorithms called “parallel tempering” and “particle filtering”. They are extremely related to it and share many features.

The authors agree with the reviewer that these (and other) approximate sampling methods are closely intertwined. However, we feel that filtering-based approaches are tangential to the present work. Filtering methods are predominately used to re-sample as new data are incorporated, either as they become available or because a large data set has been split for computational efficiency. In contrast, we study a situation where a relatively small data set is entirely known at the outset and the chief difficulty is to minimize the number of evaluations of the likelihood. For interested readers, the cited papers on TMCMC and Π 4U provide good context for TMCMC among sampling methods.

Why TMCMC and not thermodynamic integration to estimate the model evidence? How do the computational complexities differ?

As mentioned in the previous reply, computational complexity with respect to the amount of data is tangential to the main challenge of this paper. TMCMC in particular is an effective sampler for exploring a high-dimensional posterior with comparatively few evaluations of the likelihood function.

It would be interesting to also see some trace-plots of the MCMC iterations in the Appendix.

We agree with the reviewer that, given the significant differences between TMCMC and more common approximate sampling methods like Metropolis, a visual representation of convergence would be useful. We have added plots from intermediate generations to the Appendix, which we hope will provide better intuition for the reader.

The results are convincing, but I would like to also see a discussion on what would happen if, rather than considering some fixed models identified by given constrained parametrizations of the arterial model from Section 2, and then selecting between them, one just used the full arterial model. In other words: given data D from a patient with an aneurysm, it would be interesting to estimate all of the model parameters (for the full arterial network model) using TMCMC and see if the estimated values identify the aneurysm in the right location, without recurring to model selection. This would be a way to solve the problem of not having the right model in the class of models to be compared and missing a potential clinical diagnosis, with clear health risks for the patient. We should keep in mind that we want to reduce the probability of False negatives!

The reviewer makes an excellent point with respect to Type II error. Indeed, limiting comparison to a selection of models runs the risk that no model is an accurate description of the true defect. But the full arterial model is impossible because of its high dimensionality: given the complexity of the forward problem, it would be several orders of magnitude beyond practicality to sample from, e.g., a 40-dimensional posterior. (Sampling just two points in each dimension is already 2^{40} , far beyond anything attempted in this paper.) The main result of our work is a mapping of this problem to a model selection problem between much lower-dimensional models. A deeper examination of the middle ground, or alternative approaches for dealing with this dimensionality problem, would be excellent avenues for future research but are beyond the scope of the present work.

Of course the starting arterial model could be completely wrong itself, because of the structural assumptions, but in order to account for the mismatch between the model and the data there is a

literature on calibration of computer models and learning the “discrepancy term”. Perhaps a reference on this could be useful, as very likely the assumptions of the model will not hold for real data. [citation] Kennedy, M., and O’Hagan, A. (2001). Bayesian Calibration of Computer Models. Journal of the Royal Statistical Society. Series B (Statistical Methodology), 63(3), 425-464. Retrieved from <http://www.jstor.org/stable/2680584>

We thank the reviewer for this reference. As per the previous reply, this work primarily focuses on the mapping of the high-dimensional inverse problem to a low-dimensional model selection problem, so a precise treatment of model error is tangential (though of great interest for future work). However, the authors agree that further context on calibration of mathematical models would enhance the Discussion section, which has been amended accordingly.

Reproducibility and code of the results section: The code needs to be available on GitHub / Bitbucket / Gitlab, for reproducibility, along with the simulation settings and corresponding random number generator seeds.

Appendix: In the appendix, a link to the Pi4U repository would be helpful for the reader.

We thank the reviewer for identifying this oversight. We have added a link to the Pi4U repository in the Appendix.

[reference needed] Ching, J., Chen, Y.-C., 2007. Transitional Markov chain Monte Carlo method for Bayesian model updating, model class selection, and model averaging. Journal of Engineering Mechanics 133 (7), 816832.

[if any of their adjustments are used?] Betz W., Papaioannou I., Straub D. (2016): Transitional Markov chain Monte Carlo: Observations and improvements. Journal of Engineering Mechanics, ASCE, 142(5): 04016016.

The suggested reference is already cited in Section 3.3. Pursuant to the discussion in a previous reply, this has been made more clear by emphasizing the distinction between TMCMC (Ching and Chen) and Pi4U (the original implementation with efficient parallel task sharing).

Reviewer #2

The authors state (p. 27) that they introduce ‘our recent Bayesian framework for uncertainty quantification which is amenable to [dealing with noise and model misspecification].

As far as I can tell, the basic framework itself is standard Bayesian inference for inverse problems, and is not in itself ‘novel’, while the sampling method is more novel. Perhaps this is what the authors mean by ‘framework’ (e.g. the parallel computing aspect)?

Regardless, this is absolutely fine, but I would emphasise that the basic Bayesian framework is standard while the particular computational aspect is more ‘novel’ (rather than just saying ‘our’ framework) and add some citations to standard references on Bayesian inverse problems at this point e.g.

- Stuart, A. M. (2010). Inverse problems: a Bayesian perspective. Acta numerica, 19, 451-559.*
- Kaipio, J., and Somersalo, E. (2006). Statistical and computational inverse problems (Vol. 160). Springer Science and Business Media.*
- Tarantola, A. (2005). Inverse problem theory and methods for model parameter estimation (Vol. 89). SIAM.*

or other preferred/standard references.

The authors agree that the distinction between method and implementation is not sufficiently clear; this concern should be largely addressed by changes made in response to similar points by Reviewer #1. We have also incorporated some of Reviewer #2’s suggested sources (here and in what follows) and thank him/her for an excellent review of the literature in Bayesian inverse problems and robustness. This additional context should better frame the distinction between TCMC and Π 4U.

Robustness to misspecification

Regarding the ‘robustness to misspecification’ aspect, I would be more tentative. As shown in e.g. the Kaipio and Somersalo book mentioned above and related work, model misspecification can have a large effect on Bayesian inference. This is obviously the case for other inverse problems/statistical inference methods too.

In the Kaipio and Somersalo reference and related work the authors explicitly construct approximation error models to account for/guard against this misspecification. There are a number of other articles along these lines, i.e. trying to explicitly deal with misspecification, that the authors do not appear to engage with. This usually means some form of ‘wrong’ model is used, more severe than just a parameter being slightly the wrong value but the correct equations used.

*As far as I can tell, in contrast to the above, in the present work the authors simply ‘ignore’ misspecification and hope for the best, i.e. they do not explicitly account for misspecification during the inference procedures or framework. Which again, is often fine! Depending on how sensitive a particular quantity of interest is with respect to errors in other ‘nuisance’ factors. In this present work this is checked via synthetic computational case studies, which again is fine, but does not provide especially clear guidelines for when this does and does not work - there should certainly be cases where misspecification really does matter, e.g. a completely qualitatively different model is used. The authors do give one case where the ‘misspecification’ matters a bit. I would like to see more explicit and extreme examples where the misspecification really *does* matter, and more guidelines for, or discussion of, when this might happen (and why).*

*The main implication of this is that *I would recommend replacing some of the more general claims of robustness and accuracy of the model selection process by more cautious discussions in terms of ‘sensitivity studies’* (of particular quantities with respect to errors in some particular quantities).*

Finally, as indicated above, model misspecification more commonly refers to when the model structure itself is misspecified rather than simply setting one or more of the other parameters in the correct equations to (slightly) ‘wrong’ values. Have the authors considered this case? E.g. are there competing models, or models with the wrong branching structure etc which could be used to represent more systematic misspecification? Some discussion or example in this direction would greatly add to the paper.

Alternatively, I think the article framing should shift the emphasis of the robustness aspect to be about (relatively limited) ‘sensitivity’ studies rather than more general claims concerning ‘robustness’ to (true) ‘misspecification’.

In the above section, Reviewer #2 begins a very detailed look into common issues affecting Bayesian inverse methods and model selection. In particular, misspecification and robustness are generally very complex issues, and the reviewer suggests several good avenues for beginning to investigate these issues in the context of the TMCMC approach to locating arterial defects. The heart of this criticism appears in the final paragraph above: “model misspecification... refers to when the model structure itself is misspecified.” In this sense, we do not engage “true” model misspecification at all in the present work. We agree with Reviewer #2 that we fundamentally “ignore misspecification and hope for the best.”

There are two specific claims embedded in this criticism: first, that what we actually present in the paper as misspecification is really just a slight perturbation of a single model and is not a sound basis for any claim of robustness, and second, that an investigation of true model misspecification is essential to a complete presentation of the present method. To address these claims, it is important to reconsider the principal goals of the paper and its relation to the literature. There are two main novel contributions:

1. Π_4U , the parallel implementation by author P.K. et al. which is essential for practical implementation of TMCMC when evaluation of the likelihood requires running a comparatively expensive hemodynamic PDE model.
2. When attempting to locate a defect, the mapping of an impossibly high-dimensional inverse problem to a model selection problem between much lower-dimensional inverse problems which can reasonably be approximately sampled using 1.

We emphasize that model selection, as it appears in the paper, it *not* model selection between, e.g., different arterial models (either in simulation method or in network structure). Model selection in this context is simply a tool used to avoid an impossible problem by instead solving several possible problems. A key point is that the necessary calculations for model selection are already intermediate steps of TMCMC, and so no additional computational expense is incurred.

This mapping (via model selection) is itself an approximation. In order to reduce dimensionality, each model fixes the vast majority of free parameters corresponding to physical properties of the individual arterial segments. By definition, this means fixing unknown parameter values which must be estimated from data. Since estimates have error, a fundamental question is whether the proposed approach breaks down in the presence of noise in “fixed” parameters. Thus, any claim of robustness is with respect to the mapping itself, i.e., the central thrust of the present work, which leverages model selection as a useful tool. Certainly, the authors agree that we have no basis for claiming robustness to true model misspecification, since we do not consider model selection between physically distinct models of the arterial network at any point in the paper.

As Reviewer #2 points out with wonderful detail and a great number of relevant sources, claiming robustness for general model selection is an extremely challenging problem that would require investigating many additional scenarios beyond what are currently considered in this paper. We also agree that robustness is of central importance for practical clinical implementation of the approach. Nonetheless, the authors legitimately feel that a complete treatment of robustness to general misspecification would go far beyond the core goal of this paper (establishing model selection as a practical way to “solve” an extremely high-dimensional Bayesian inverse problem which is very expensive to sample). As such, these issues must be left for future work.

We acknowledge that this discussion is a valuable one. We have amended several sections of the paper (most notably the Discussion) to better convey the points we have stressed in this response. In particular, we have been careful to frame our claim of robustness in terms of error in model parameters. We are sincerely thankful to Reviewer #2 for raising these issues.

Minor Comments

– Line 31/31. Say ‘Equation (1)’ rather than starting the sentence with (1).

Changed as suggested. We thank the reviewer for this small note.

– In section 3.3. the authors should reference their PI4U article(s). Some discussion of/references to/comparison to other sampling methods used in large-scale Bayesian inverse problems would also be useful.

These valid criticisms are now addressed by revisions made in response to previous comments.

– Data availability. As far as I can see the authors code and data are not available in a manner such that other people could run and reproduce their results. I think the authors should make enough code/data available (e.g. via GitHub or similar) such that someone could easily reproduce at least some portion of this analysis (while perhaps being limited by available parallel computing facilities).

We thank Reviewer #2 for noting this oversight as well. While the code was already freely available, this was not made clear anywhere in the paper. The repository is now linked in the Appendix.

On behalf of the authors,
Anastasios Matzavinos